# Evaluation of Antibody Tests for *Mycobacterium bovis* Infection in Pigs and Deer

**DOI:** 10.3390/vetsci10080489

**Published:** 2023-07-27

**Authors:** Penny Barton, Nick Robinson, Sonya Middleton, Amanda O’Brien, John Clarke, Maria Dominguez, Steve Gillgan, John Selmes, Shelley Rhodes

**Affiliations:** 1Animal and Plant Health Agency (APHA), Weybridge KT15 3NB, UK; penny.barton@apha.gov.uk (P.B.); nicholas.robinson@apha.gov.uk (N.R.); sonya.middleton@apha.gov.uk (S.M.); maria.dominguez@apha.gov.uk (M.D.); stephen.gillgan@apha.gov.uk (S.G.); john.selmes@apha.gov.uk (J.S.); 2Enfer Scientific, W91 FD74 Naas, County Kildare, Ireland; amandaobrien@enfergroup.com (A.O.); johnclarke@enfergroup.com (J.C.)

**Keywords:** antibody, diagnostic, ELISA, lateral flow, IDEXX, Enferplex, DPP VetTB, *Mycobacterium bovis*, pig, deer

## Abstract

**Simple Summary:**

This study describes the evaluation of four antibody tests (three ELISA tests and one lateral flow) for the detection of *Mycobacterium bovis* infection in deer and pigs. Test sensitivity and specificity were derived for each test with associated test cut-offs. There was a high level of test agreement between the tests. High test specificity was achieved, with a high to moderate test sensitivity, depending on whether or not a prior skin test had been performed, respectively. These data allow for the confident use of antibody tests for pigs and deer in GB, where previously none were available.

**Abstract:**

This study addressed the need in Great Britain for supplementary blood tests for deer and pig herds under movement restrictions due to confirmed *Mycobacterium bovis* infection—to enhance the overall sensitivity and reliability of tuberculosis (TB) testing and contribute to an exit strategy for these herds. We evaluated four antibody tests (lateral flow DPP VetTB Assay for Cervids, *M. bovis* IDEXX ELISA, Enferplex Cervid and Porcine antibody tests and an in-house comparative PPD ELISA) using serum samples from defined cohorts of TB-infected and TB-free deer and pigs. TB-infected deer included two separate cohorts; farmed deer that had received a tuberculin skin test less than 30 days prior, and park deer that had received no prior skin test. In this way, we were able to assess the effect of the skin test anamnestic boost upon antibody test sensitivity. We tested a total of 402 TB-free pigs and 416 TB-free deer, 77 infected farmed deer and 105 infected park deer, and 29 infected pigs (including 2 wild boar). For deer, we found an equivalent high performance of all four tests: specificity range 98.8–99.5% and sensitivity range 76.6–85.7% for skin test-boosted infected deer, and 51.4–58.1% for non-boosted infected deer. These data suggest an overall approximate 25% increase in test sensitivity for infected deer following a skin test boost. For pigs, the tests again had equivalent high specificity of 99–99.5% and a sensitivity range of 62.1–86.2%, with substantial agreement for three of the four tests. Retrospective application of the ELISA tests to individual culled park deer and wild boar that showed no obvious evidence of TB at larder inspection identified a significant seropositivity within wild boar suggestive of low-level *M. bovis* infection that would otherwise not have been detected. Overall this investigation provided a robust evaluation of four antibody tests, which is essential to generate confidence in test performance before a wider deployment within TB control measures can be considered.

## 1. Introduction

There is currently no routine statutory surveillance testing program for bovine tuberculosis (TB) in deer or pig herds in Great Britain (GB), which have largely been regarded as spill-over hosts for *Mycobacterium bovis* [1]. The passive surveillance of animals, generally at slaughterhouse meat inspection, has been considered sufficient to identify lesions suspicious of TB, and such tissues are submitted to the Animal and Plant Health Agency (APHA) for mycobacterial speciation. Historically, this has involved mycobacterial culture to identify *M. bovis*, which can take six or more weeks. From 30 March 2022, APHA began testing suspect lesions from non-bovines using a validated polymerase chain reaction (PCR) test to allow for more rapid identification of *M. bovis* and implementation of TB control measures. However, meat inspection is not a sensitive tool, e.g., estimates of detecting *M. bovis* in pigs have been stated as 25% in fattening pigs and 50% in adults (apha.defra.gov.uk/external-operations-admin/library/generics/Tuberculosis/Pigs/Case_Management.html (accessed on 19 February 2023)).

APHA non-bovine statistics for pigs and deer show the number of cases to be relatively low compared to cattle, but not insignificant. A total of 32 deer and 56 pig premises were under TB restrictions at the close of 2021. While some of these restrictions were in place as a result of overdue contiguous tests imposed because of a neighbouring (usually cattle) confirmed herd breakdown, others were due to the suspicion or confirmation of *M. bovis* infection. In 2021, new confirmed breakdowns were identified in three pig and nine deer premises, and 2202 individual deer and 4082 individual pig tests were carried out with 136 test-positive deer slaughtered as a result (there were no test-positive pigs). Suspicious lesions from 42 individual deer and 205 individual pigs were cultured in 2021, with 20 deer (48%) and 4 (2%) pigs confirmed as *M. bovis*-infected. Similarly, APHA data for 2022 (January to September) show *M. bovis*-positive results (mix of culture and PCR tests) in 36 out of 50 deer (72%), and 9 out of 184 pigs (4.9%). The relatively low identification of *M. bovis* from pig TB lesions clearly supports *M. bovis* not being the most common cause of TB-like lesions in pigs—in fact, seven cases of *M. microti* in pigs were identified in the 2022 dataset. *M. avium* continues to be suspected as the main cause of TB in pigs [1], but the culture systems utilised at APHA for TB diagnostics are optimised for the growth of *M. bovis* and as such may not be optimal for *M. avium.*

Herds affected by *M. bovis* are managed under various TB legislative Orders (Tuberculosis in Animals (England) Order 2021; Tuberculosis in Specified Animals (Scotland) Order 2015; Tuberculosis (Wales) Order 2011. However, the lack of an effective “exit strategy” for TB-restricted deer and pig herds has sometimes resulted in restrictions remaining in place indefinitely, and where there is little or no impact upon the business models of restricted farms, there is also little incentive to pro-actively engage in TB testing to lift those restrictions.

TB testing for deer and pigs remains reliant upon the comparative intradermal tuberculin skin test, which compares the response to both bovine and avian tuberculins (PPDB and PPDA, respectively) and provides a result based upon the level of PPDB-bias as a measure of *M. bovis*-specific response (and therefore *M. bovis* infection). The test is cervical for deer, and at the base of the ear for pigs—and both the deer and pig industries hold concerns regarding the practice and effectiveness of the test. The comparative skin test is not validated for either species under GB conditions, but the application of the test over many years in GB and elsewhere has generally supported confidence in its continued use. In the case of deer, the comparative skin test is thought to have high sensitivity (85.8%) and specificity (97.7%) [2], but the small margins of the test to denote a positive, negative, or (in particular) inconclusive skin reactor, has led to some mistrust in its correctly identifying infection. For pigs there is no EFSA opinion on the performance of the skin test—performance is thought to be high based upon (i) two small studies summarised by Cousins and Florisson [3] suggesting 100% specificity in 25 TB-free pigs and 17 TB-free feral pigs, and 100% sensitivity in 19 *M. africanum*-infected pigs, and (ii) APHA statistics that support a high specificity, with a very low overall test-positivity in pigs during 2020 and 2021 (respectively 2 positives out of 4076 tested, and no positives out of 4082 tested). For both pigs and deer, there are perceived difficulties in the practicalities of the skin test, which, in the case of pigs, often leads owners to depopulate their herds rather than test. Uncertainties over the practicality and usefulness of the skin test for deer and pigs have led to studies of supplementary blood tests that, when used together with the skin test could increase the sensitivity and/or specificity of infection detection. The need for blood-based tests for pigs and deer in GB was also recognised in the 2018 Bovine TB Strategy Review [4].

This study aimed to evaluate antibody tests for pigs and deer that could be made available in the short to medium term within the APHA test portfolio, providing a practical contribution to an effective and acceptable “exit strategy” for deer and pig herds under TB restriction. The tests we investigated were the lateral flow DPP VetTB assay for CervidsTM (Chembio Diagnostic Systems, Medford, NY, USA), an in-house comparative PPD ELISA, Enferplex Cervid and Porcine TB antibody tests (Enfer Scientific Laboratories Inc., Nass, Ireland) and the IDEXX *M. bovis* ELISA (Idexx Laboratories Inc., Westbrook, ME, USA)—all of which had published credentials for TB testing in pigs or deer.

We tested defined cohorts of TB-infected and TB-free deer and pigs with all four antibody tests and used Receiver Operator Curve (ROC) analyses to provide test sensitivity, specificity, and optimal test cut-offs. Our data suggest a strong performance on all tests for pigs and deer, with a high level of test agreement for most tests. We were further able to quantify, for deer, the increase in test sensitivity afforded by anamnestic skin test boosting, and show the usefulness of antibody testing for estimates of seroprevalence in wild deer and boar populations.

## 2. Materials and Methods

### 2.1. Animals/Serum Samples

#### 2.1.1. Farmed Deer

A total of 77 serum samples from 49 skin test-positive deer and 28 skin test-negative deer with gross visible lesions (VL) of TB deer (65 Red (*Cervus elaphus*) and 12 Fallow (*Dama dama*)) from 5 recently confirmed *M. bovis*-infected premises in England (Gloucestershire, Dorset, Wiltshire, Hereford and Worcester, and Staffordshire). Deer were euthanised within 10–30 days of the prior skin test and blood samples were collected immediately post-slaughter to provide anamnestic sera. Of the 77 deer, there were 57 with VL, and VL deer were present on all premises. Mycobacterial culture on samples from 4 of the 5 premises (one farm had only one skin reactor, a VL that was not submitted for culture), returned positive *M. bovis* culture results (genotypes of 9a, 10, 9 and 21 for Cornwall, Gloucestershire, Dorset and Wiltshire, respectively).

#### 2.1.2. Farmed Pigs

A total of 27 serum samples from skin test-positive pigs (25 *Sus scrofula domesticus*—unspecified breed plus 2 *Kune kune*) from four confirmed *M. bovis*-infected premises in England (Cornwall, North Yorkshire, Shropshire and Hereford/Worcestershire). Pigs were euthanised within 10–30 days of the prior skin test and blood samples were collected immediately post-slaughter to provide anamnestic sera. Of the 27, there were 9 with VL (and VL pigs were present at each of the 4 premises), 5 of which from 3 premises returned a positive *M. bovis* culture result (genotypes of 9, 17a and 35 from Cornwall, Hereford/Worcester and Shropshire, respectively).

#### 2.1.3. Forest of Dean Wild Boar

A total of 237 blood samples were collected from 243 boar (*Sus scrofa*) culled by the Forestry Commission in the Forest of Dean (FoD) between September 2019 and March 2020 as part of boar population control. Blood samples were collected from shot boar at larder inspection. Boar found with VL were submitted to APHA for culture. Two VL boar were identified, and both returned an *M. bovis* culture-positive, genotype 9.

#### 2.1.4. Park and Wild Deer

A total of 105 samples from park and wild VL deer (37 Red (*Cervus elaphus*) and 68 Fallow (*Dama dama*)) were used in this study—blood samples were taken from shot deer at larder inspection. There were 98 VL from 302 deer culled on three separate deer park premises (Worcestershire, Gloucestershire, and Somerset) between September 2019 and March 2020, of which 76% were *M. bovis* culture-confirmed (genotypes as follows: Worcestershire 17, 9 and 10, Gloucestershire 17, Somerset 11). Infection was confirmed across male and female deer, and red and fallow were present on the same premises. Seven additional samples were added to this cohort from *M. bovis* culture-positive VL wild deer culled between October 2007 and January 2008 as part of a previous study [5].

#### 2.1.5. TB-Free Farmed Deer

A total of 416 samples (from 429) collected immediately post-slaughter by Dovecote Park Ltd. (Pontefract, UK), Yorkshire were used in this study. Samples used originated from a total of 13 holdings identified from low-risk areas of England or Scotland and with no history of TB within the past five years. All were red deer yearlings, 70% male, 30% female, all NVL at meat inspection.

#### 2.1.6. TB-Free Farmed Pigs

A total of 402 serum samples were identified from the APHA Surveillance and Laboratory Service Division annual pig survey. Samples used originated from 14 holdings with no TB history within the past five years. Most samples were derived from the Low-Risk Area of England and from Scotland but some were also from the Edge and High-Risk areas of England. No information was available regarding the age or sex of the pigs.

Serum samples were prepared from a single blood sample from each animal and stored at −20 °C. Aliquots were then prepared for testing on all four antibody tests. Owing to some samples being of low volume, not all samples could be tested on all tests, e.g., for TB-free deer 416 samples were tested using IDEXX and DPP VetTB tests, but only 410 of these were tested using Enferplex and PPD ELISA tests.

### 2.2. Antibody Tests

#### 2.2.1. IDEXX ELISA

The commercial IDEXX *M. bovis* cattle antibody test (#99-29853, Idexx Laboratories, Westbrook, ME, USA), was modified for use in non-bovines by replacing the kit secondary detection anti-bovine-HRP (peroxidase) reagent with either Protein-G-HRP (Sigma-Aldrich, St. Louis, MU, USA #P8170) for deer or Protein-A/G-HRP (Thermo Fisher Scientific, Waltham, MA, USA #32490) for pigs—since Protein-G-HRP did not show good sensitivity for pigs in this test. ELISA plates pre-coated with a cocktail of the immuno-dominant antigens MPB83 and MPB70 were used to test up to 92 serum samples/plates in single wells. Serum samples were diluted 1/50 using the kit sample buffer and 100 μL of diluted sample was loaded onto the ELISA plate and incubated for one hour at room temperature (RT). Plates were washed 6× using kit wash buffer and 100 μL of the secondary detection reagent diluted 1/20,000 in 1%BSA/PBS was added and plates were incubated for 30 min at RT. Plates were washed and developed for 15 min using the kit developing and stop solutions and the Optical Density (OD) was read on an ELISA Reader (range 0–6) at 450 nm (OD450 nm). Kit bovine positive and negative plate controls were included in all tests as test run quality controls (QC). Test readouts are illustrated as the OD450 nm value for each sample (animal).

#### 2.2.2. Comparative PPD (∆PPD) ELISA

NUNC MAXISORP ELISA plates were coated overnight at 4 °C with 50 μL/well of PPDA and PPDB (matched tuberculin vials used from the same kit batch, PPDB 3000 IU and 2500 IU, respectively; Prionics, Thermo Fisher Scientific) each diluted 1/2000 in carbonate coating buffer (pH9.6). Well contents were removed, and the plates were blocked using 200 μL/well of 20% soya milk/PBS. Plates were incubated for 2 h at RT. Plates were washed 6× using PBS/0.1% Tween 20, and 50 μL of serum sample, diluted 1/50 in block solution, was added to each well. Each serum sample was added to 2 wells—one PPDA-coated and one PPDB-coated. Plates were incubated for 2 h at RT, then washed as above. 50 μL/well of secondary detection reagent (as above, Protein-A/G-HRP for pigs & Protein-G-HRP for deer) was added and plates were incubated for one hour at RT. Plates were washed and 100 μL/well of substrate solution was added (TMB, #T0440-1L; [Sigma-Aldrich], Merck KGaA, Darmstadt, Germany) for 5 min at RT, and the reaction stopped with 100 μL/well 0.5 M sulphuric acid. Plates were read on an ELISA reader (range 0–6) at OD450 nm, and the *M. bovis*-specific response for each sample was determined by subtracting the PPDA_OD450nm_ from the PPDB_OD450nm_ (∆OD450 nm PPDB-PPDA). Test data are illustrated as the ∆OD450 nm (PPDB-PPDA) for each sample (animal). Different batches of PPD were investigated for potential batch variation in the ∆PPD ELISA, but no evidence for this was found—all batches gave identical test readouts.

#### 2.2.3. Enferplex Cervid TB Assay and Enferplex Porcine TB Assay

The Enferplex tests were carried out at the Enfer Scientific laboratory in Ireland. A proprietary collection of 11 antigens consisting of soluble products, recombinant proteins either singly or in mixtures or as fusion proteins and synthetic peptides were selected based on reactions detected using positive and negative sera from animals known to be free from or infected with *M. bovis*. The antigens were deposited in a multiplex planar array as individual 30nl spots into wells of 96-well black polystyrene microtiter plates using BioDot aspirate/dispense platforms. Plates were blocked, stabilised, dried, and stored at 2–8 °C until use.

For the Enferplex Cervid TB assay, antigens were as follows; Rv2975 synthetic peptide p6; PPDb; recombinant Rv2873; recombinant Rv2975; Bovine TB cocktail; recombinant Rv3875-Rv3874 fusion protein; recombinant Rv3874-Rv3875 fusion protein; recombinant Rv3616c; recombinant Rv3881c; recombinant Rv3803c; recombinant Rv1860. Serum samples were diluted 1:150 into sample dilution buffer (Enfer Buffer F, Enfer Scientific, Newhall, Ireland) and mixed before 50 μL was added per well. The plates were incubated at 37 °C for 60 min with agitation (900 rpm). The plates were washed 6 times with 1X Wash (Enfer Wash buffer, Enfer Scientific) and aspirated. The detection antibody, chicken anti-cervid IgG—HRP (Gallus Immunotech, Fergus, ON, Canada) was prepared as a 1:12,000 dilution.

For the Enferplex Porcine TB assay, antigens were as follows; Rv2975 synthetic peptide p6; PPDB; recombinant Rv2873; recombinant Rv2975; Bovine TB cocktail; recombinant Rv2031c-Rv1886c fusion protein; recombinant Rv3875-Rv3874 fusion protein; recombinant Rv3874-Rv3875 fusion protein; recombinant Rv3616c; recombinant Rv3881c; recombinant Rv2031c. Serum samples were diluted 1:200 into sample dilution buffer (Enfer Buffer G, Enfer Scientific) and mixed before 50 μL was added per well. The plates were incubated at 37 °C for 60 min with agitation (900 rpm). The plates were washed 6 times with 1X Wash (Enfer Wash buffer, Enfer Scientific) and aspirated. The detection antibody, goat anti-pig IgG—HRP (Novus Biologicals) was prepared as a 1:25,000 dilution.

For both assays, the plates were incubated at 37 °C for 60 min with agitation (900 rpm). The plates were washed as above and 50 μL of prepared chemiluminescent substrate (50:50 dilution of substrate and diluent), (Advansta Inc., San Jose, CA 95131, USA) was added per well. Relative light units (RLU) were captured (220 s exposure) immediately, using Quansys Biosciences Q-ViewTM LS imager and Q-ViewTM software (v 3.12). The results for each sample were defined using the Enferplex Cervid TB Macro and the Enferplex Porcine TB Macro, based on individual antigen thresholds after subtracting the RLU value obtained from a blank spot. A positive test was defined as a sample for which 2 spots or more showed positive responses on either the High Sensitivity or High Specificity test readout. Test data are illustrated as the number of spots positive for each sample (animal).

#### 2.2.4. DPP VetTB Lateral Flow

The DPP^®^ VetTB Assay for Cervids is a USDA-approved commercial test for cervids in the USA and as such is optimised for cervids as a qualitative (visual) test at overall high sensitivity (74.3%) and specificity (97.5%) [6,7,8]. The test incorporates 2 antigen lines; T1 (MPB83) and T2 (ESAT6/CFP10), and the detection reagent is a Protein-A/G-colloidal gold conjugate. As well as a visual test, the cassette may be inserted into an optical cassette reader (Optricon Reader, Chembio Diagnostic Systems, Inc., Medford, NY, USA), which provides a quantitative (numerical) readout that may be used to optimise the test for other species, as is currently done for camelid testing at APHA. Optricon readings for both T1 and T2 are recorded as Relative Light Units (RLU) for each sample. Test data are illustrated as the RLU for T1 and T2 for each sample (animal).

### 2.3. Statistical Analyses

Test readouts were analysed with Receiver Operator Curve (ROC) statistical analysis (GraphPad Prism 8). This provides the estimates for test sensitivity and specificity with associated 95% confidence intervals (95%CI) at a given test cut-off. Agreement between the different tests for pigs and for deer was investigated using (i) McNemars 2-tailed test for paired samples with a *p* value < 0.05 suggesting a significant difference between tests, and also (ii) Kappa (k) statistic—which takes into account chance agreement, thus when two measurements agree only at the chance level, the value of kappa is zero, with perfect agreement at 1.0. The kappa scale of agreement is as follows; 0.0–0.2 slight; 0.21–0.4 fair; 0.41–0.6 moderate; 0.61–0.8 substantial; 0.81–1.0 almost perfect.

## 3. Results

### 3.1. Antibody Tests Performance—Sensitivity and Specificity—In Deer and Pigs

Antibody tests for deer: Figure 1 shows all deer antibody test data for the Enferplex, IDEXX and ∆PPD ELISA tests plus the DPP VetTB lateral flow test as individual T1 and T2 antigens. Each symbol represents one deer sample in each test. The farmed skin reactor deer and park/wild VL deer are shown in red, and the TB-free deer are shown in green. ROC analyses provided the cut-offs for each of the IDEXX, ∆PPD ELISA and the DPP T1 and T2 antigens as follows; IDEXX > 0.502; ∆PPD ELISA B-A > 0.43; DPP Vet TB must have a visible response at T1 or T2 (associates with T1 or T2 > 60RLU). The Enferplex test cut-off was provided by Enfer Scientific at ≥ 2 spots for a positive result. There was little or no difference in the Enferplex readouts for High Sensitivity or High Specificity therefore the High Specificity data only are shown in Figure 1. A summary of overall test performances for deer is provided in Table 1. All tests showed a range of test readouts for the seropositive TB-infected farmed and park deer. Notably, the Enferplex test for skin test-boosted farmed deer contained individuals with some of the highest spots-positive compared to the unboosted park group. Similarly, responses to the DPP VetTB antigen T2 were higher in the skin test-boosted farmed deer compared to T2 responses in the unboosted park group.

Statistical comparison of the deer tests (Table 2) suggested an equivalence of high specificity across all four tests (98.8–99.5%). The sensitivity of all tests was higher in the farmed deer (76.6–85.7%) than in the park/wild deer (51.4–58.1%). This was likely due to the anamnestic effect of the skin test in farmed deer enhancing antibody test sensitivity in that group, which was not present in park/wild deer. Test comparisons for the 182 TB-infected deer (77 farmed and 105 park/wild) using the kappa statistic and McNemar’s test are shown in Table 2. The kappa statistic for deer tests shows “substantial” to “almost perfect” agreement on all test comparisons. The standard error (SE) for kappa is minimal, and the 95% confidence interval (95%CI) ranges for kappa are also mostly “substantial” to “almost-perfect”. McNemar’s test comparisons for deer tests did suggest a significant difference for two test comparisons—between the Enferplex and ∆PPD ELISA, and between the Enferplex and the IDEXX ELISA. This likely reflects that in farmed deer the Enferplex test identified a slightly higher percentage of infected deer (85.7% for Enferplex compared to 79.2% for ∆PPD ELISA and 76.6% for IDEXX ELISA)—the 95% CI for all were overlapping, however, with Enferplex achieving a slightly higher upper limit compared to the other 2 tests.

Antibody tests for pigs: Figure 2 shows all antibody test data for the Enferplex, IDEXX and ∆PPD ELISA tests plus the DPP VetTB lateral flow test for individual antigens T1 and T2. Each spot represents one pig in each test. The farmed skin reactor pigs (TB-infected group) are shown in red, and the TB-free pigs are shown in green. The two infected wild boar are denoted by solid red symbols. ROC analyses provided the cut-offs for each of the IDEXX, ∆PPD ELISA and the DPP T1 and T2 antigens as follows; IDEXX > 0.76; ∆PPD ELISA B-A > 0.65; DPP VetTB must have a visible response at T1 or T2 (associates with T1 or T2 > 100 RLU). The Enferplex test cut-off was provided by Enfer Scientific at ≥ 2 spots positive. There was little/no difference in Enferplex readouts for Enferplex High Sensitivity or High Specificity therefore the data for High Specificity only are shown in Figure 2. A summary of individual test performances is provided in Table 3. While the specificities of all four tests were high, there was some suggested difference in test sensitivities, with the ∆PPD ELISA appearing to have relatively lower sensitivity compared to the other tests. As for deer tests, TB-infected pigs also displayed a range in the strength of responses across all tests—with the two wild boar providing some of the highest responses observed.

Statistical test comparisons for the 29 TB-infected pigs using the kappa statistic and McNemar’s test are shown in Table 4. The kappa statistic for pig test comparisons showed “substantial” agreement between IDEXX, DPP VetTB and ∆PPD ELISA tests. However, due to the small cohort size the SE and 95%CI around the kappa statistic estimates were higher than those for deer. McNemar’s test comparison for pigs did show a significant difference between the ∆PPD ELISA and both the DPP VetTB and Enferplex tests—again likely reflecting the lower sensitivity of the ∆PPD ELISA (62.1%) compared to DPP VetTB and Enferplex tests (both 86.2%) in this small cohort of TB-infected pigs.

Agreement of seropositive pig and deer identification across tests. Table 5 summarises the numbers of TB-infected seropositive pigs and deer that were tested with all four tests and were positive for at least one test. There was no difference between farmed and park deer in terms of the agreement, so all 135 seropositives are grouped together. The data show that for deer, 74.8% of seropositives were identified by all four tests, and for pigs, 66.7% of seropositives were identified by all four tests, with reducing numbers of individuals thereafter identified by three, two and then just one test. These data support the fact that the tests identify the same infected individuals in most cases, with some outliers positive for one test but not another. The data appear relatively skewed in pigs (compared to deer) with 5 seropositives (18.5%) identified by only one of the four tests (in this case 3 Enferplex-positive only and 2 DPP VetTB-positive only), again likely due to the low number of TB-infected pigs in this study.

Combining antibody tests was investigated for potential improvement in sensitivity or specificity. For this exercise, each test was paired with every other test, and a combined test result was generated using a parallel interpretation for higher sensitivity (either test may be positive for an overall positive result), and a serial interpretation for higher specificity (both tests must be positive for an overall positive result) (see Appendix A for combined test data: Appendix A—Pig parallel tests; Appendix A—Pig Serial Tests; Appendix A—Deer Parallel Tests; Appendix A—Deer Serial Tests). Table 6 suggests that a small but not insignificant increase in test sensitivity of infection (of >7%) could be achieved by parallel testing where individuals had received a prior skin test anamnestic boost. This effect was lower (3.9%) in infected deer that had not received a skin test. The reduction of specificity as a result of parallel testing was ~0.6% for pigs and deer. These data, while crude averages across tests, suggest a significant benefit of parallel combined testing over single test application in confirmed infected herds where individuals have received a prior skin test. Table 7 suggests minimal benefits of combining tests in a serial interpretation, with minimal increases in specificity (0.5% for pigs and 0.87% for deer) and associated decreases in test sensitivity—6.5% for deer and 8.9% for pigs. N.B. the loss of sensitivity in deer not skin tested was lower at 3.6%.

### 3.2. Seroprevalence Estimates among Wild Boar within the Forest of Dean and Park/Wild Deer with No Visible Lesions of TB

We assessed the seropositivity of individual shot wild boar and deer that were found to have no visible lesions (NVL). Importantly, larder inspection involved the whole body, not eviscerated carcase, inspection. The three ELISA tests (IDEXX, Enferplex and ∆PPD) were used to screen serum samples from all NVL park deer and wild boar, to investigate the potential for *M. bovis* infection being present, perhaps at a low level, but missed by larder inspection.

Table 8 shows the percentage of seropositives identified within 233 wild boar and 197 deer for each of the three ELISA tests. Data from deer NVLs suggested that little or no infection was being missed during larder inspection—seropositivities for all three tests were low and approximating the false-positive (95%CI) ranges for the tests. In contrast, data from wild boar NVLs suggested that significant numbers of infected individuals were being missed at larder inspection—seropositivities were in excess of the false-positive ranges for the tests. Collectively the test data for wild boar in the FoD suggest a seropositivity of ~15%.

## 4. Discussion

This study investigated the performances of four antibody tests (IDEXX, Enferplex and ∆PPD ELISAs and the DPP VetTB lateral flow test) that could practically be made available within APHA in the short to medium term to aid the diagnosis of *M. bovis* infection in pigs and deer. The results suggest a high performance for all tests in terms of sensitivity and specificity and a high level of agreement across tests in terms of detecting infected individuals. There were some small differences, notably with pigs—likely due to the small TB-infected group size. However, even with pigs, there was a high specificity (≥99%) for all tests and an associated high sensitivity (72.4–86.2%) in three of the four tests.

The larger numbers of TB-infected deer allowed for higher confidence in the test performance estimates provided by the ROC analyses. There was a good level of agreement between all tests; with a moderate sensitivity in the identification of TB-infected deerpark/wild deer (51–58%) and a higher sensitivity in the identification of TB-infected farmed deer (76.6–85.7%)—all at high specificity (98.8–99.5%). The difference in sensitivity between park/wild and farmed deer was likely due to the benefit of the skin test applied to farmed deer—the data from this study quantified the anamnestic boost effect in infected deer as ~25%. The higher test sensitivity afforded to farmed deer following a skin test supports the application of supplementary antibody tests in confirmed breakdown herds to detect infected individuals that are skin-test-negative. In our TB-infected farmed cohort, for e.g., there were 28 such seropositive individuals that were VL and *M. bovis* culture-positive. Even in the absence of a prior skin test, the lower antibody test sensitivities may be effective in the identification of infected farmed deer for which a skin test is not possible, e.g., stags that are difficult to handle, and/or to assess seroprevalence on deer parks or other geographical areas [5] where infection is known or suspected to be present. While most deer in this study were either red or fallow, we were able to test an additional small group of cervids comprising 5 Reindeer, 9 Sika and 1 Axis deer—all from TB-free premises and all test-negative on the 4 antibody tests.

Our test performances accord well with the few published studies that have used these tests for TB detection in pigs and deer. Supplementary use of the DPP VetTB test with the skin test on fallow deer (*Dama dama*) in Spain was shown to increase the overall sensitivity of infection detection from 76% (skin test alone) to 97% (skin test followed by the DPP VetTB test) [9]. The DPP VetTB test was also used by Busch et al. [10] in a large, infected red deer (*Cervus elaphus*) herd in England—a study that demonstrated successive rounds of the skin test with supplementary antibody testing for skin test-negative and inconclusive skin reactors could clear TB. More recently, the test has been applied to culled red deer on Exmoor to assess seroprevalence [11]. The DPP VetTB test was described as having a sensitivity of 61.5–69.2% in wild boar piglets (*Sus scrofa*) in an endemic region of Spain [12], and 75% sensitivity in South African warthogs (*Phacochoerus africanus*) [13]. The DPP VetTB, being essentially a non-species-specific test, is amenable across species where the options of qualitative (by eye) or quantitative (by numerical Optricon cut-off) readout may be applied. APHA uses a validated DPP VetTB test for use in camelids and badgers [14,15].

In New Zealand the ETB (ELISA TB) test, essentially a comparative PPD antibody ELISA, is used as a serial test in deer to add specificity to single intradermal skin test (SIT)-positive reactors—the SIT having a higher sensitivity but lower specificity relative to the comparative skin test. It has been a key test in the drastic reduction of infected deer between 2005 and 2021 [16]. The ETB also formed part of the test regime used by Busch et al. [10] above.

Various Enferplex antibody tests (Enfer Scientific, Ireland) have been successfully applied to cattle [10,17,18,19], goats [20] and camelids [15], and here we also now describe a high performance of that test for pigs and deer. The IDEXX *M. bovis* ELISA, a commercial test for TB in cattle modified for non-bovine (camelid) use at APHA [15] is now, with different test cut-offs, also of use for pigs and deer.

We used the ELISA tests evaluated for sensitivity and specificity in this study to assess the seropositivity of culled deer and wild boar that were NVL on larder inspection. The ELISA data suggested that (i) the whole body inspection of culled deer in this project resulted in very little infection being missed (very low seropositivity in NVL deer), while (ii) in wild boar, infection is likely being missed (significant seropositivity in NVL boar), possibly because *M. bovis* infection is very low level and/or better controlled immunologically in the boar. The pig Enferplex test cut-offs originally having been set using wild boar samples suggests this porcine assay is as relevant to wild boar as domestic pig samples, and provides further confidence of these estimates of seroprevalence in the FoD boar.

Unlike for deer tests, we had no opportunity to assess pig test sensitivity in the absence of a prior skin test. If, like deer tests, the sensitivity is significantly lower without a skin test, then the ~15% seroprevalence in FoD will be an underestimate of the actual prevalence of infection. Wild boars constitute a potential TB risk for domestic pig herds in endemic areas of England where both co-exist—this is underscored by a recent study [21] near the Forest of Dean that described the visitations of wild boar to two commercial pig farms. Elsewhere in Europe wild boar are considered to be significant maintenance hosts of *M. bovis* and a source of infection for cattle [12,22,23,24,25], while in South Africa *M. bovis* infection in warthogs risks disease spread within and between national parks and wildlife reserves [13]. In New Zealand, the susceptibility of feral pigs has led to their use as sentinel hosts to monitor the prevalence of *M. bovis* in the brushtail possum wildlife reservoir [26]. In GB, while pigs are considered mostly as incidental spill-over hosts for *M. bovis*, the gross pathology described in one small study [27] described thoracic, abdominal and generalised lesions, from which arises the potential for onward transmission via either pulmonary or faecal spread.

## 5. Conclusions

The results in this study highlight four main findings: (1) A high and equivalent test performance, particularly for deer, across the four different antibody tests. Any of these antibody tests may, therefore, be used with confidence to detect *M. bovis* infection, thus providing flexibility in test choice to match scenario, test availability and workstream. Indeed, deer antibody tests are now being validated for use by APHA. For pigs, three of the four tests showed strong equivalent test performances; however, the TB-infected pig cohort was small, and it is hoped that further accumulation of samples over time will allow for a more accurate evaluation of test performances. (2) Quantification of the beneficial anamnestic effect of a prior skin test upon antibody test sensitivity—for deer, this was ~25%, demonstrating the clear benefit of supplementary antibody testing with the skin test wherever possible. (3) Combined parallel antibody testing for both pigs and deer in infected herds can be used to enhance antibody test sensitivity where a prior skin test has been performed. (4) Our data show that robustly evaluated tests of defined performance, even with moderate sensitivity, can provide useful estimates of seroprevalence, and so exposure to *M. bovis*. Test evaluation data such as this study describes for pigs and deer in GB is essential for the consideration of test introduction as part of the TB control program. Antibody tests represent a proportionate, relatively simple and inexpensive way of improving the diagnosis of TB in non-bovine species, where numbers affected are significant but remain relatively small.

## Figures and Tables

**Figure 1 vetsci-10-00489-f001:**
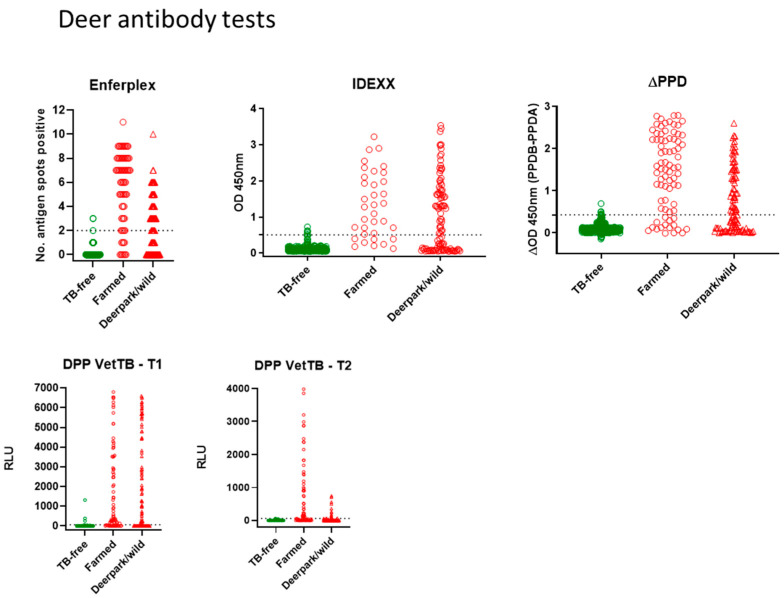
Antibody tests for deer. Test readouts are shown for each test (Enferplex number of antigen spots-positive, IDEXX OD450 nm, ∆PPD [OD450_PPDB_-OD450_PPDA_] and DPP VetTB T1 and T2 Relative Light Units [RLU]). Each spot represents one animal (*n* = 77 farmed and *n* = 105 park deer for all tests; *n* = 410 TB-free deer for Enferplex and ∆PPD tests; *n* = 416 TB-free for IDEXX and DPP VetTB tests). The cut-offs (horizontal dotted lines) were as follows: IDEXX > 0.502; ∆PPD ELISA B-A > 0.43; DPP Vet TB must have a visible response at T1 or T2 (associates with T1 or T2 > 60RLU); Enferplex provided by Enfer Scientific at 2 spots or more positive. There was little/no difference in the Enferplex readouts for High Sensitivity or High Specificity therefore the High Specificity data only are shown.

**Figure 2 vetsci-10-00489-f002:**
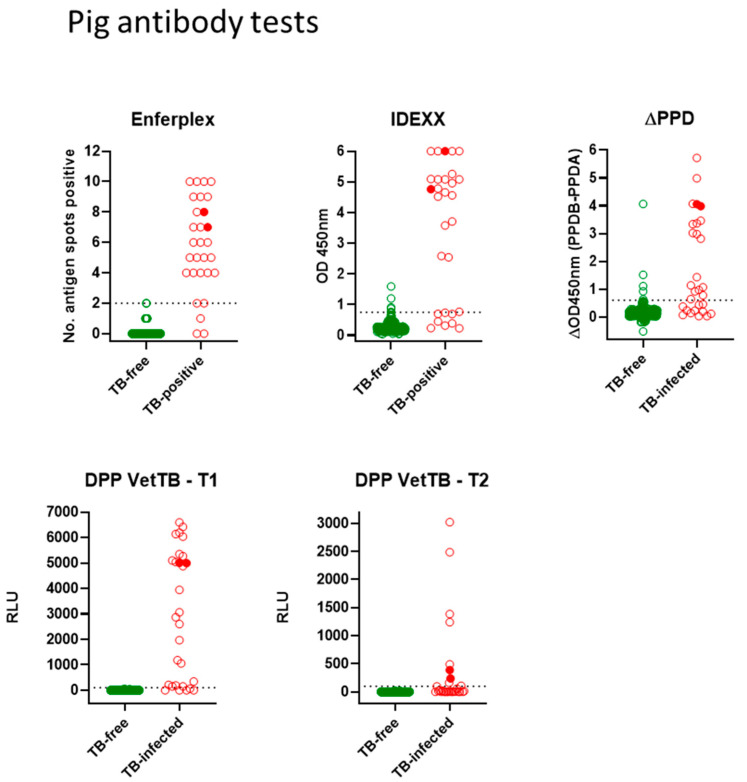
Antibody tests for pigs. Test readouts are shown for each test (Enferplex number of antigen spots-positive, IDEXX OD450 nm, ∆PPD [OD450_PPDB_-OD450_PPDA_] and DPP VetTB T1 and T2 Relative Light Units [RLU]). Each spot represents one animal (*n* = 29 TB-infected pigs and wild boar for all tests; *n* = 400 TB-free pigs for Enferplex and ∆PPD tests; *n* = 402 TB-free for IDEXX and DPP VetTB tests). The two infected wild boar are denoted by solid red symbols. The cut-offs (horizontal dotted lines) were as follows: IDEXX > 0.76; ∆PPD ELISA B-A > 0.65; DPP VetTB must have a visible response at T1 or T2 (associates with T1 or T2 > 100 RLU); Enferplex test cut-off was provided by Enfer Scientific at 2 spots or more positive. There was little/no difference in Enferplex readouts for Enferplex High Sensitivity or High Specificity therefore the data for High Specificity only are shown.

**Table 1 vetsci-10-00489-t001:** Summary of deer antibody test performance. Sensitivity (Se) and specificity (Sp) of each antibody test are shown, derived from statistical analyses of the test data shown in Figure 1. The test cut-offs to achieve these performances were as follows; IDEXX > 0.502; ∆PPD ELISA B-A > 0.43; DPP Vet TB must have a visible response at T1 or T2 (associated with T1 or T2 > 60RLU); Enferplex test cut-off was provided by Enfer Scientific at 2 spots or more positive.

	TB-Free	TB-Infected	
	% Specificity [95%CI]	% Sensitivity [95%CI]	
	(No Prior Skin Test)	Park/Wild: VL/Mb+ (No Prior Skin Test)	Farmed: Skin Test+/VL/Mb+ (with Prior Skin Test)
IDEXX ELISA	98.8 [97.2–99.6]	55.2 [45.2–65.0]	76.6 [65.6–85.5]
DPP VetTB	99.0 [98.1–100]	58.1 [48.7–67.5]	77.9 [68.6–87.2]
∆PPD ELISA	98.8 [97.2–99.6]	51.4 [41.5–61.3]	79.2 [68.5–87.6]
Enferplex High Se	99.0 [98.1–100]	56.2 [46.7–66.7]	85.7 [77.9–93.5]
Enferplex High Sp	99.5 [ 98.8–100]	55.2 [45.7–64.7]	85.7 [77.9–93.5]

**Table 2 vetsci-10-00489-t002:** Statistical comparison of deer antibody tests. Agreement between the different tests for deer was investigated using (i) Kappa (k) statistic, and (ii) McNemars 2-tailed test for paired samples, with a *p* value < 0.05 suggesting a significant difference between tests.

TB-Positive Deer	Kappa Statistic				McNemar
	*k* Value	Level of Agreement	SE of *k*	95%CI	*p* Value
IDEXX v. DPPD ELISA	0.83	almost perfect	0.043	0.75–0.92	0.789
IDEXX v. DPP VetTB	0.81	substantial	0.046	0.72–0.90	0.453
Enferplex v. ∆PPD ELISA	0.81	substantial	0.046	0.70–0.90	0.024
IDEXX v. Enferplex	0.78	substantial	0.049	0.68–0.88	0.01
DPP VetTB v. Enferplex	0.72	substantial	0.055	0.62–0.83	0.522
DPP VetTB v. ∆PPD ELISA	0.69	substantial	0.056	0.58–0.80	0.327

**Table 3 vetsci-10-00489-t003:** Summary of pig antibody test performance. Sensitivity (Se) and specificity (Sp) of each antibody test are shown, derived from statistical analyses of the test data shown in Figure 2. The test cut-offs to achieve these performances were as follows; IDEXX > 0.76; ∆PPD ELISA B-A > 0.65; DPP VetTB must have a visible response at T1 or T2 (associated with T1 or T2 > 100 RLU). The Enferplex test cut-off was provided by Enfer Scientific at 2 spots or more positive.

Pig Tests	TB-Free (No Prior Skin Test)	TB-Infected Skin Test+/VL/Mb+ (with Prior Skin Test)
	% Specificity [95%CI]	% Sensitivity [95%CI]
IDEXX ELISA	99.0 [97.5–99.6]	72.4 [54.3–85.3]
DPP VetTB	100.0 [99.1–100]	86.2 [73.6–98.8]
∆PPD ELISA	99.0 [97.5–99.6]	62.1 [44.0–77.3]
Enferplex High Se	99.3 [98.5–100]	86.2 [73.6–98.8]
Enferplex High Sp	99.5 [ 98.8–100]	86.2 [73.6–98.8]

**Table 4 vetsci-10-00489-t004:** Statistical comparison of pig antibody tests. Agreement between the different tests for deer was investigated using (i) Kappa (k) statistic, and (ii) McNemars 2-tailed test for paired samples, with a *p* value < 0.05 suggesting a significant difference between tests.

TB-Positive Pigs	Kappa Statistic				McNemar
	*k* Value	Level of Agreement	SE of K	95%CI	*p* Value
IDEXX v. ∆PPD ELISA	0.77	substantial	0.123	0.53–1.0	0.248
IDEXX v. DPP VetTB	0.71	substantial	0.153	0.41–1.0	0.248
IDEXX v. Enferplex	0.59	moderate	0.173	0.25–0.93	0.134
DPP VetTB v. ∆PPD ELISA	0.51	moderate	0.156	0.20–0.81	0.041
Enferplex v. ∆PPD ELISA	0.42	moderate	0.156	0.11–0.72	0.023
DPP VetTB v. Enferplex	0.34	fair	0.231	−0.11–0.8	1.00

**Table 5 vetsci-10-00489-t005:** Agreement of seropositive pigs and deer across tests. The numbers and proportion of TB-infected seropositive pigs and deer that were tested with all four tests and were positive to at least one test are shown.

	Pigs (*n* = 27)		Deer (*n* = 135)	
Number of Tests Positive (out of 4)	*n*	%	*n*	%
4	18	66.7	101	74.8
3	3	11.1	15	15
2	1	3.7	10	10
1	5	18.5	9	9

**Table 6 vetsci-10-00489-t006:** Parallel combined antibody testing for higher sensitivity. Mean sensitivity and specificity estimates are shown for single antibody tests (4 options) versus the various combinations of two tests (6 options) interpreted in parallel (see Appendix A for parallel test combination data). The % increase in sensitivity of parallel combined testing relative to single test application is shown, plus the associated % decrease in test specificity.

		Mean Test % Sensitivity			Mean Test % Specificity		
		Single	Parallel	% Increase Se	Single	Parallel	% Decrease Sp
Deer	with prior skin test	79.9	87.2	7.3			
	no prior skin test	55.0	58.9	3.9	99.03	98.4	0.63
Pig		76.7	83.9	7.2	99.4	98.8	0.6

**Table 7 vetsci-10-00489-t007:** Serial combined antibody testing for higher specificity. Mean sensitivity and specificity estimates are shown for single antibody tests (4 options) versus the various combinations of two tests (6 options) interpreted in serial (see Appendix A for serial test combination data). The % increase in specificity of serial combined testing relative to single test application is shown, plus the associated % decrease in test sensitivity.

		Mean Test % Specificity			Mean Test % Sensitivity		
		Single	Serial	% Increase Sp	Single	Serial	% Decrease Se
Deer	with prior skin test				79.9	73.4	6.5
	no prior skin test	99.03	99.9	0.87	55.0	51.4	3.6
Pig		99.4	99.9	0.5	76.7	67.8	8.9

**Table 8 vetsci-10-00489-t008:** ELISA seropositivity of NVL wild boar and park deer. The specificity (and 95%CI) is shown for each ELISA test together with the proportion (%) of seropositives found among the wild boar and park deer with no visible lesions (NVL) on larder inspection.

	Wild Boar (*n* = 233)		Park Deer (*n* = 197)	
ELISA Test	% Sp [95%CI]	% Positive	% Sp [95%CI]	% Positive
IDEXX	99.0 [97.5–99.6]	17.2	98.8 [97.2–99.6]	3.6
∆PPD	99.0 [97.5–99.6]	15.4	98.8 [97.2–99.6]	5.1
Enferplex—High Se	99.3 [98.5–100]	19.7	99.0 [98.1–100]	2.5
Enferplex—High Sp	99.5 [98.8–100]	19.3	99.5 [98.8–100]	2.5

## Data Availability

The (anonymised) data used in this study are available from the corresponding author upon reasonable request.

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
