# Peer review of "Evaluation of Antibody Tests for Mycobacterium bovis Infection in Pigs and Deer"

_vetsci, 2023, doi:10.3390/vetsci10080489_

Round 1

Reviewer 1 Report

The paper describes the evaluation of serological tests for tuberculosis in pigs, wild boars and deer. The authors performed extensive research. The presentation of the results is very good and detailed. Methodically, the research is correct. However there is lack of great novelty.  

The title is misleading, it seems that it may only be about domestic pig, not pig and boar. Maybe it's worth adding Latin names. The Introduction seems too long and some of the information should be transferred to the Discussion, which is actually missing in the article. Similarly, there are no conclusions in the abstract. Simple summary is not posted at all. In addition, the paper contains many shortcomings, such as punctuation errors (eg.  120) or lack of italics when using Latin names (e.g. 61, 62, 113, 116, 122, 123, 576). The Introduction does not end with a goal as is usually assumed, but rather with the conclusion of the research in this paper. The paper structure is incorrect, and the biggest shortcoming is the lack of Discussion. The final chapter should be expanded and titled Discussion, not Conclusion and expanded significantly. Some information to Discussion can be transferred from the Introduction, some are lacking (more comparisons to other species, previous studies needed).

Unnecessary information was left in sections such as founding, institutional review board statement and others.

The authors declare no conflict of interest, however, one of the authors works for the company that produces the enfraplex test.

Author Response

Author Response to Reviewer 1:

Thank you for your considered review of our manuscript. I realise how time-consuming this can be, and fully appreciate your time and thoughtful comments, which I have acted upon as below (in blue type).

With best wishes,

Shelley Rhodes.

  1. The paper describes the evaluation of serological tests for tuberculosis in pigs, wild boars and deer. The authors performed extensive research. The presentation of the results is very good and detailed. Methodically, the research is correct. However there is lack of great novelty.  

This is foremost a paper of practical usefulness – as we stated, it is the evaluation of tests that could be made available for use in GB in the short to medium term where no validated tests currently exist, and where a need for such tests has been expressed at government level.

What this MS underscores following a process of robust evaluation in cohorts of defined TB infection status animals is; (i) First -  test equivalence - providing flexibility in antibody test choice to match scenario, test availability and workstream. (ii) Second - this paper quantifies the value of the anamnestic skin test boost to enhance antibody test sensitivity, (iii) Third, this paper describes the benefit of parallel combined antibody testing to enhance overall sensitivity of infection detection (but only where a prior skin test boost has been given), and (iv) Finally, application these well-evaluated tests in populations of unknown infection status (culled deer and boar) provides confidence in the seroprevalence estimates so generated.

  1. The title is misleading, it seems that it may only be about domestic pig, not pig and boar. Maybe it's worth adding Latin names.

It was felt this would lengthen the title rather a lot. The word “domestic” was not mentioned in the title as we wanted to encompass all pigs (commercial breeds and boar) and all deer (here mainly red and fallow). But we take your point that this detail was lacking and have added the deer and pig species within 2.1. Animals / Serum Samples.

  1. The Introduction seems too long and some of the information should be transferred to the Discussion, which is actually missing in the article.

Agreed. After re-checking author guidance, “4. Conclusions” has been changed to “4. Discussion”, with an “In conclusion” paragraph added to underscore the main messages of the paper (as written above).

Text describing the tests that was originally in the Introduction has been moved to a more appropriate position in the Discussion section of the paper. Also some minor text has been deleted throughout where messages were not altered.

  1. Similarly, there are no conclusions in the abstract. Simple summary is not posted at all.

Agree – concluding comments now added to the abstract. A Simple Summary is now provided.

  1. In addition, the paper contains many shortcomings, such as punctuation errors (eg.  120) or lack of italics when using Latin names (e.g. 61, 62, 113, 116, 122, 123, 576).

Apologies – consistent use of italics has been corrected (another Reviewer also spotted this!).

I have checked for punctuation errors.

  1. The Introduction does not end with a goal as is usually assumed, but rather with the conclusion of the research in this paper.

I have checked again the author format guidance – It does say to “Finally, briefly mention the main aim of the work and highlight the main conclusions.”

I have therefore re-worked the end of the Introduction to state:

The aim of the study - lines 94-97

The tools (tests) we used - lines 97-101

What we tested, and what we found - lines 102-108

  1. The paper structure is incorrect, and the biggest shortcoming is the lack of Discussion.

See above, now corrected.

  1. The final chapter should be expanded and titled Discussion, not Conclusion and expanded significantly. See above, now corrected.

  1. Some information to Discussion can be transferred from the Introduction, some are lacking (more comparisons to other species, previous studies needed

See above, now corrected. Regarding different species, they are mentioned where they concern the antibody tests pertinent to this MS (see Discussion 477-78 badgers and camelids; 486-487 goats and cattle). We wanted to focus on the tests being evaluated and on pigs and deer, rather than widen into a review of TB serology.

  1. Unnecessary information was left in sections such as founding, institutional review board statement and others. These are editorial inclusions/requests that I have now dealt with.

  1. The authors declare no conflict of interest, however, one of the authors works for the company that produces the enfraplex test. Actually two co-authors are Enfer Scientific employees, but samples were provided blinded as to their infection status, and test data analysis of unblinded samples was carried out at APHA.

Reviewer 2 Report

The article addresses an important topic, especially for developed countries where health problems, such as animal tuberculosis, are already under control or even eradicated. And adoption of different diagnostic methods to eliminate possible animals or identify susceptible species are important and can be replicated in other countries. However, many serological tests, mainly ELISA, need to be evaluated and validated with greater repeatability and in a longer period of time, so that they can be indicated as possible methods to replace, for example, the intradermal test with PPD bovis and avium. The manuscript, as a complete article, has some serious flaws from the structural point of view. There is neither a Simple Summary nor a discussion of the data, with others available in the literature. Remembering that for an article the discussion is the most important point. In addition, the references do not have DOI access, which makes it difficult for the reviewer to read the referenced articles and assess their scientific merit. And finally, the figures are not didactic and do not add scientific merit to the work, and the tables are out of format. From my humble point of view, the manuscript needs to be rewritten in the communication format and the rules of the journal must be respected from the point of view of scientific structure. It is not simply taking a lot of good data and putting it on paper in a descriptive way, if it is necessary to discuss the data, without discussion this document is a mere report of an experiment.

Author Response

Author Response to Reviewer 2:

Thank you for your considered review of our manuscript. I realise how time-consuming this can be, and fully appreciate your time and thoughtful comments, which I have acted upon as below (in blue type).

With best wishes,

Shelley Rhodes.

The manuscript, as a complete article, has some serious flaws from the structural point of view:

  1. There is neither a Simple Summary nor a discussion of the data, with others available in the literature.

I did not see the request for a Simple Summary in the Author guidance, apologies. This was identified also by another Reviewer, and is now provided.

  1. Remembering that for an article the discussion is the most important point.

I have changed the Conclusion section to a Discussion, and added an “In conclusion” paragraph at the end of the Discussion section.

I have moved text/references that discuss the antibody tests out of the Introduction and into a more appropriate place in the Discussion section.

  1. In addition, the references do not have DOI access, which makes it difficult for the reviewer to read the referenced articles and assess their scientific merit. I re-checked author guidance and it stated that “DOI numbers are not mandatory but highly encouraged” DOI numbers were provided where available.

  1. And finally, the figures are not didactic and do not add scientific merit to the work, and the tables are out of format.

Figures 1 & 2 show the test data in their entirety, rather than tables which summarise ROC analyses of that data. Therefore we believe the figures do add value, and are not repetitive of other information in the MS.

Tables 1 and 3 are now re-formatted and so are now amenable to editorial change – which I believe was applied to all other tables apart from these two.

  1. From my humble point of view, the manuscript needs to be rewritten in the communication format

I do not agree with this paper being shortened to a Communication. Nor has this this been mooted by any other Reviewer.

  1. and the rules of the journal must be respected from the point of view of scientific structure. It is not simply taking a lot of good data and putting it on paper in a descriptive way, if it is necessary to discuss the data, without discussion this document is a mere report of an experiment. Please see above - restructuring has been carried out as a result of this peer review.

Reviewer 3 Report

Review of manuscript, vetsci-2407538, “Evaluation of antibody tests for Mycobacterium bovis infection in pigs and deer” by Barton et al.

This manuscript describes the use of 4 different antibody tests on various populations of deer and pigs. The introduction provides the reader with sufficient background to understand the problem and the objectives of the study. The authors adequately describe the antibody tests, the study populations, as well as the results. The authors conclude that the tests performed well in terms of sensitivity and specificity and there was a high level of agreement between tests. There was an approximate 25% increase in sensitivity in farmed deer that had been previously skin tested, confirming the known boosting effect of PPD administration. This information is useful to animal health officials involved in bovine tuberculosis eradication efforts.

I have only minor typographical errors to point out.

Genus and species are not italicized in many cases. 

The abbreviation for microliters is often @l.

Author Response

Author Response to Reviewer 3:

Thank you for your considered review of our manuscript. I realise how time-consuming this can be, and fully appreciate your time and feedback, which I have acted upon as below (in blue type).

With best wishes,

Shelley Rhodes

I have only minor typographical errors to point out.

  1. Genus and species are not italicized in many cases. 

Apologies for lack of some italics – now corrected .

  1. The abbreviation for microliters is often @l.

Greek symbols appear to have been altered in the version Reviewers received – I have now corrected these.

Round 2

Reviewer 1 Report

All comments have been taken into account by the Authors. 

Author Response

Dear Reviewer,

Thank you so much for your time in reviewing our MS, and providing helpful comments to improve our narrative. It is very much appreciated.

with best wishes,

Shelley Rhodes (on behalf of all authors)

Reviewer 2 Report

I acknowledge and thank the authors for their effort in making adjustments to the original article, with the aim of improving its quality, now presenting a manuscript with topics such as Simple Summary; Discussion; formatting tables and improving the spelling of some terms of scientific nomenclature; Funding; Informed Consent Statement; Data Availability Statement and even an improvement of bibliographic references.

However, some problems are still observed:

1). The conclusion topic does not exist now, it must be inserted.

2). The way of writing the Author Contributions does not conform to the journal, as the names must be abbreviated. For example: Conceptualization, Shelley Rhodes - Conceptualization, S.R. (these are rules, the authors actually used the template from the Veterinary Sciences?). Fix all Author Contributions.

3). Institutional Review Board Statement. As an article involving the use of animals, animal restraint, animal sacrifice ("Pigs were euthanized within 10-30 days of prior skin test and blood samples collected at slaughter to provide anamnestic sera" lines 134-136), collection of biological materials from animals do not have the evaluation and authorization of an ethics committee? That's not possible. Please submit a document from an institution with an Ethics Committee on Animal Experimentation that evaluated the project and approved its realization, or a very well-founded justification to justify "This study did not require ethical approval". It’s in the Veterinary Sciences template “The animal study protocol was ap-proved by the Institutional Review Board (or Ethics Committee) of NAME OF INSTITUTE (protocol code XXX and date of approval).” for studies involving animals."

4). References. Despite a change in most of the references, including mainly articles that enabled a better construction of the Discussion topic, of the 27 references, only 3 have DOI access, which once again makes it difficult for the reviewer to assess the content and quality of the references. And of the same 27 new references, 14 (approximately 50%) have been published for more than 10 years, and reference 16 does not fit a specific citation pattern ("16. New Zealand Annual Report for Bovine Tuberculosis: 2022-Surveillance-Annual -Report-No-49-3 September-Animals-TB-pdf"). Here again the authors do not follow the Veterinary Sciences template. The formatting of the references is totally wrong, the year of publication is in the wrong place, the year is not in bold, the names of the scientific journals are not in italics. Example: 14. Ashford, R.T., Anderson, P., Waring, L., Dave, D., Smith, F., Delahay, R.J., Eamonn Gormley, E., Chambers, M.A., Sawyer, J. & S. Lesellier . 2020. Evaluation of the Dual Path Platform (DPP) VetTB assay for the detection of Mycobacterium bovis infection in badgers. Preventive Veterinary Medicine 180: 105005 (https://doi.org/10.1016/j.prevetmed.2020.105005).

Correct would be Ashford, R.T., Anderson, P., Waring, L., Dave, D., Smith, F., Delahay, R.J., Eamonn Gormley, E., Chambers, M.A., Sawyer, J. & S. Lesellier. Evaluation of the Dual Path Platform (DPP) VetTB assay for the detection of Mycobacterium bovis infection in badgers. Prev. Vet. Med. 2020, 180, 105005. (https://doi.org/10.1016/j.prevetmed.2020.105005).

These small but big details reveal a certain disregard on the part of the authors in reviewing the publication, since it is the magazine's rule to follow the Template and you state that "Writing – review & editing, Nick Robinson, Sonya Middleton, Amanda O'Brien, John Clarke, Maria Dominguez, Steve Gillgan, John Selmes and Shelley Rhodes", of the 9 authors of the article 8 did the review and editing of the article, unfortunately it does not seem that this really happened with attention.

5). The authors' response "From my humble point of view, the manuscript needs to be rewritten in the communication format. I do not agree with this paper being shortened to a Communication. Nor has this this been mooted by any other Reviewer." The fact that another reviewer does not think like the other is what makes it possible to improve the work, here you should convince me and justify why a manuscript should not be a communication, but a original article and not just saying I don't agree, my opinion remains.

An article is formed by the quality of the information, the quality of the discussions and, above all, following the journal's standards, especially with regard to structuring and formatting. This is fundamental for the journal to have a standard and be considered a quality journal and not a predatory journal that accepts an article in any form. Reviewers, editors and authors must ensure the credibility of the journal Veterinary Sciences and the MDPI group.

Author Response

Author Response to Reviewer 2 (2)

I acknowledge and thank the authors for their effort in making adjustments to the original article, with the aim of improving its quality, now presenting a manuscript with topics such as Simple Summary; Discussion; formatting tables and improving the spelling of some terms of scientific nomenclature; Funding; Informed Consent Statement; Data Availability Statement and even an improvement of bibliographic references.

However, some problems are still observed:

1). The conclusion topic does not exist now, it must be inserted.

As stated in my previous response – we were a little confused over whether the final section should be “Discussion” or “Conclusion”. This was changed to “Discussion” after peer review (and also re-worked as requested). I also added an “In conclusion” paragraph at the close of the Discussion section, to bring together the major points from this MS.

I feel that our response has been sufficient.

We did find conflicting author instructions on the journal webpages – one instruction stating “research manuscripts should comprise: Introduction, Materials & Methods, Results, Discussion, Conclusion (optional)”, while the Free Format submission guidance states Introduction Materials & Methods, Results and Conclusion…(?)

2). The way of writing the Author Contributions does not conform to the journal, as the names must be abbreviated. For example: Conceptualization, Shelley Rhodes - Conceptualization, S.R. (these are rules, the authors actually used the template from the Veterinary Sciences?). Fix all Author Contributions.

Author contributions were provided via the journal webpage drop-down menu of contributions for each co-author. I had no control over how this information was displayed in the final MS.

3). Institutional Review Board Statement. As an article involving the use of animals, animal restraint, animal sacrifice ("Pigs were euthanized within 10-30 days of prior skin test and blood samples collected at slaughter to provide anamnestic sera" lines 134-136), collection of biological materials from animals do not have the evaluation and authorization of an ethics committee? That's not possible. Please submit a document from an institution with an Ethics Committee on Animal Experimentation that evaluated the project and approved its realization, or a very well-founded justification to justify "This study did not require ethical approval". It’s in the Veterinary Sciences template “The animal study protocol was ap-proved by the Institutional Review Board (or Ethics Committee) of NAME OF INSTITUTE (protocol code XXX and date of approval).” for studies involving animals."

Ethics committee was not relevant to this work, as I have already stated elsewhere for this submission, because there was no blood sampling of live animals. All samples were either (i) opportunistic blood samples taken after the animals were culled (TB-infected deer and pigs, wild boar, TB-free deer) or (ii) samples already within the APHA collection having been collected for other surveillance (non-TB) reasons (TB-free pigs). I have now updated “2. Materials & Methods, 2.1. Animals / serum samples” to clarify any potential misunderstanding on this issue (lines 125, 135, 143, 148, 157).

4). References. Despite a change in most of the references, including mainly articles that enabled a better construction of the Discussion topic, of the 27 references, only 3 have DOI access, which once again makes it difficult for the reviewer to assess the content and quality of the references.

As I raised previously, Veterinary Sciences author guidance states that DOI numbers are not mandatory (Free format Submission guidelines below). Where they were available, they were added.

“Free Format Submission

Veterinary Sciences now accepts free format submission:

  • We do not have strict formatting requirements, but all manuscripts must contain the required sections: Author Information, Abstract, Keywords, Introduction, Materials & Methods, Results, Conclusions, Figures and Tables with Captions, Funding Information, Author Contributions, Conflict of Interest and other Ethics Statements. Check the Journal Instructions for Authors for more details.
  • Your references may be in any style, provided that you use the consistent formatting throughout. It is essential to include author(s) name(s), journal or book title, article or chapter title (where required), year of publication, volume and issue (where appropriate) and pagination. DOI numbers (Digital Object Identifier) are not mandatory but highly encouraged. The bibliography software package EndNote, Zotero, Mendeley, Reference Manager are recommended.
  • When your manuscript reaches the revision stage, you will be requested to format the manuscript according to the journal guidelines.”

And of the same 27 new references, 14 (approximately 50%) have been published for more than 10 years,  

The references used were relevant to this work.

and reference 16 does not fit a specific citation pattern ("16. New Zealand Annual Report for Bovine Tuberculosis: 2022-Surveillance-Annual -Report-No-49-3 September-Animals-TB-pdf"). Here again the authors do not follow the Veterinary Sciences template.

A link for the NZ report is now supplied – apologies.

A link is also now provided for the EFSA Journal Opinion, 2008 article.

The formatting of the references is totally wrong, the year of publication is in the wrong place, the year is not in bold, the names of the scientific journals are not in italics. Example: 14. Ashford, R.T., Anderson, P., Waring, L., Dave, D., Smith, F., Delahay, R.J., Eamonn Gormley, E., Chambers, M.A., Sawyer, J. & S. Lesellier . 2020. Evaluation of the Dual Path Platform (DPP) VetTB assay for the detection of Mycobacterium bovis infection in badgers. Preventive Veterinary Medicine 180: 105005 (https://doi.org/10.1016/j.prevetmed.2020.105005). Correct would be Ashford, R.T., Anderson, P., Waring, L., Dave, D., Smith, F., Delahay, R.J., Eamonn Gormley, E., Chambers, M.A., Sawyer, J. & S. Lesellier. Evaluation of the Dual Path Platform (DPP) VetTB assay for the detection of Mycobacterium bovis infection in badgers. Prev. Vet. Med. 2020, 180, 105005. (https://doi.org/10.1016/j.prevetmed.2020.105005). These small but big details reveal a certain disregard on the part of the authors in reviewing the publication, since it is the magazine's rule to follow the Template and you state that "Writing – review & editing, Nick Robinson, Sonya Middleton, Amanda O'Brien, John Clarke, Maria Dominguez, Steve 2 Gillgan, John Selmes and Shelley Rhodes", of the 9 authors of the article 8 did the review and editing of the article, unfortunately it does not seem that this really happened with attention

The author instructions state that; “Your references may be in any style, provided that you use the consistent formatting..” and we did try to be consistent. The instruction also says that “When your manuscript reaches the revision stage, you will be requested to format the manuscript according to the journal guidelines.” We were asked by the Editor to number all references, which was then done. However as further change has now been requested here, all references have been re-formatted as per this latest request.

5). The authors' response "From my humble point of view, the manuscript needs to be rewritten in the communication format. I do not agree with this paper being shortened to a Communication. Nor has this this been mooted by any other Reviewer." The fact that another reviewer does not think like the other is what makes it possible to improve the work, here you should convince me and justify why a manuscript should not be a communication, but a original article and not just saying I don't agree, my opinion remains.

Communication generally refers to short reports with limited data.

Our MS contains a quantity of new test data that we believe other diagnostic scientists would find interesting and practically useful, and which requires full explanation and display in a full-sized article – hence this MS.

While I absolutely agree with the individuality of review, nevertheless two of three Reviewers have had no issues with either the size or content of this MS.

Therefore I can only leave this to an Editorial decision.

An article is formed by the quality of the information, the quality of the discussions and, above all, following the journal's standards, especially with regard to structuring and formatting. This is fundamental for the journal to have a standard and be considered a quality journal and not a predatory journal that accepts an article in any form. Reviewers, editors and authors must ensure the credibility of the journal Veterinary Sciences and the MDPI group

Our MS submission followed the author guidance as provided.

We took particular note of the Free Format instruction, this being an online journal, which states “no strict formatting requirements”  (though to include certain sections – which we followed – with a bit of confusion over whether there should be a “Discussion” or “Conclusion” section). With References, the author guidance states “any style, provided consistent”.

This being our first submission to this journal, we were not familiar with the format and found the instructions/guidance to be a little in conflict, but did the best we could, and certainly willing to alter any of these aspects following peer review. Which have been done.

We certainly stand by the quality of new information provided in our MS. We present original data of practical test application which responds to a requirement for diagnostic TB testing in these non-bovines.  There appears to be otherwise Reviewer agreement that the work is well designed (with sufficient, relevant background information), with the methods and results clearly presented and described, and conclusions supported by the data. This provides for robust tests that can now be easily and confidently reproduced in any other veterinary diagnostic laboratory.

We have updated our MS following Reviewer comments and do feel that it has been improved as a result. And we have responded to format change requests, which we hope now meet with the requirements of this journal.